# Cooperative Design of Devices and Services to Balance Low Power and User Experience †

Takayuki Hoshino [1,2,*], Rentaro Yoshioka [2,*], Yukihide Kohira [2] and Shingo Tetsuka [2]

1   Nihon Unisys, Ltd., Technology Research & Innovation, Tokyo 135-8350, Japan
2   Graduate School of Computer Science and Engineering, University of Aizu, Fukushima 965-8580, Japan; kohira@u-aizu.ac.jp (Y.K.); m5251147@u-aizu.ac.jp (S.T.)
*   Correspondence: takayuki.hoshino@biprogy.com (T.H.); rentaro@u-aizu.ac.jp (R.Y.)
†   This paper is an extended version of our paper published in the IEEE 14th International Symposium on Embedded Multicore/Many-core Systems-on-Chip. Hoshino, T.; Yoshioka, R.; Kohira, Y. Design of a Knowledge Experience Based Environment for Museum Data Exploration and Knowledge Creation, 2021 IEEE 14th International Symposium on Embedded Multicore/Many-core Systems-on-Chip (MCSoC), 2021, pp. 296–303, doi:10.1109/MCSoC51149.2021.00051.

**Abstract:** CPS (Cyber Physical Systems) is an approach often adopted for improving real-world activities by utilizing data. It also can be used to improve customer experiences in service applications by analyzing customer behavior, captured by sensing devices and by supporting utilization of that data by the service providers, to improve the system. In developing such systems, no method has been established to systematically evaluate the impact of individual component design on the user experience. Knowledge Experience Design is a method for distilling and validating information that affects the quality of the user experience by focusing on user activities and underlying knowledge. This methodology has been applied to a system for a museum, in which visitor activities are observed by sensing devices, to aid the Curator's awareness for improving museum services. As a result, a cooperative process for designing devices and user experience as a service was derived, in which competing interests of lower power consumption and user experience improvement have been attained. The proposed design method can be used for the co-design of systems that are built on the close coordination of hardware devices and software applications, for providing value-oriented services to users, which aids realization of CPS oriented to evaluating and improving such environments.

**Keywords:** Cyber Physical System; low power; co-design; user experience

## 1. Introduction

CPS (Cyber Physical Systems) is effective for utilizing data and improving the quality of real-world activities. Adoption of CPS is largely encouraged in the manufacturing industry, but it can also be effective in the service sector. The Smart Museum Project is an initiative to realize a system to monitor and improve museum services by sensing the behavior of visitors with passive sensing devices, and analyzing and inferring the viewing experience of visitors [1]. The system incorporates Knowledge Experience Design, a design methodology that focuses on user activities (user experiences) and the underlying knowledge, and encourages design decisions that best preserve such knowledge that is useful in maximizing the service value. This paper presents an application of Knowledge Experience Design to a system involving co-design of a hardware device and application service that are interdependent, which accomplished power reduction in the device, while preserving the quality of the user experience that is often sacrificed in preference of efficiency and performance.

To highlight user experiences in service design, a service can be modeled using a Front Stage and a Back Stage analogy, as depicted in Figure 1 [2,3]. The Front Stage represents

the experience of the end users of that service, whereas the Back Stage represents the experience of the providers of that service, who will observe the current status, consider improvements, and apply necessary control. In the Back Stage, the context-sensitive control of the Front Stage is generated by a process called Observe-Orient-Decide-Action [4,5]. First, the situation of the Front Stage is observed through collected data. Then, an orientation that best responds to the situation is formed, which leads to a decision, and the corresponding action is executed as a control. In carrying out this process, it is necessary to utilize (learn) knowledge that consists of skills, rules, knowledge, and expertise [6,7].

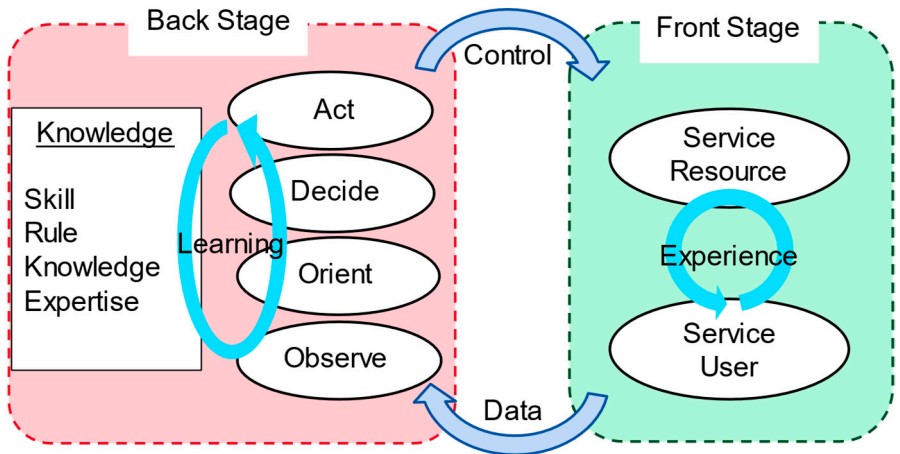

**Figure 1.** Front Stage–Back Stage view of a service.

By designing an information system based on this model, the collected real-world (Front Stage) data is analyzed/recognized in cyberspace (Back Stage) and corresponding knowledge is accumulated. Controlling real-world services based on accumulated knowledge improves the quality of the user experience, just as in a CPS.

It is one of the roles of the museum to analyze and understand the visitor experience, improve the exhibition service, and improve the experience of visitors [8]. The model of the museum service is shown in Figure 2. The Smart Museum Project aims to acquire the experiences of visitors as data and control the exhibition based on it. In this case, the "control" may include, for example, optimizing explanations to visitor interests, adjusting the exhibit hall environment (lighting, temperature, etc.), and providing route guidance to avoid congestion.

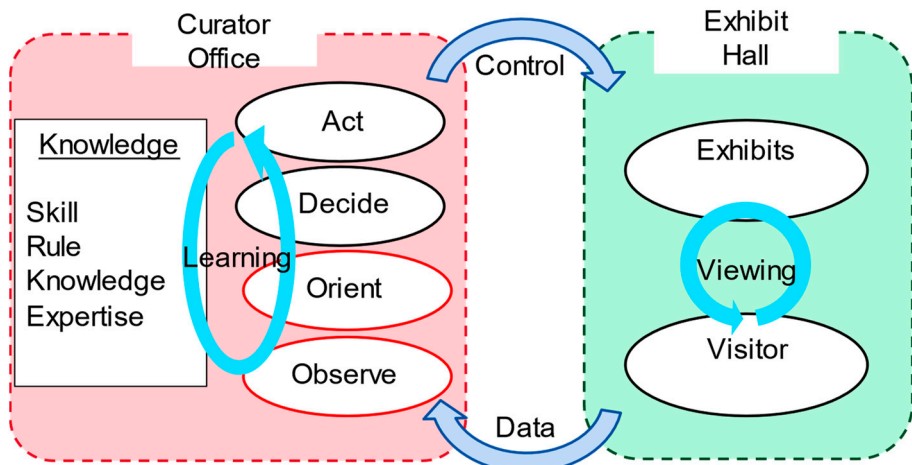

**Figure 2.** The proposed service model of Smart Museum.

To design an information system based on this model, criteria for evaluating the collected data and for deciding the control, according to each situation, play an important role. In the museum service of this study, first, it is necessary to identify data types and measurement methods that are effective in controlling the exhibition. Then, the requirements on the system for supporting the Curator's behavioral processes in the "observe" and "orient" phases of the process need to be defined. In the context of the target service, the requirements should include the support of Curators to explore and analyze the data collected in the exhibit hall and to engage in learning.

The Knowledge Experience Design is applied to the design and development of the system. Through the operation of the prototype system based on Knowledge Experience design, the accumulation of learning (knowledge) that leads to the control of the exhibition, and the types of data and measurement methods that are useful for the control of the exhibition, will be clarified.

The following describes the design and implementation of the Smart Museum prototype system based on the Knowledge Experience design. Next, the approaches related to Knowledge Experience design are introduced and the characteristics of Knowledge Experience design are shown. Then, based on the data measured by the prototype system and the knowledge obtained from the data, the type of the data and the measurement method of the data are considered. Specifically, it is shown that Knowledge Experience design contributes to the co-design of systems and services through the issue of power saving in data measurement.

## 2. Method: Design a Prototype System Based on Knowledge Experience

The prototype system needs to collect data that contributes to the accumulation of learning (knowledge) that leads to the control of the exhibition as a result of the Curator's exploratory analysis. In other words, it is necessary to design the type of data to be collected and the method of collection in coordination with the user experience of acquiring knowledge of Curators.

Hence, the Knowledge Experience Design (KED) method was applied as the basis for collaborative design focusing on user experience. Knowledge Experience is a cyclical mechanism that expresses and shares the user experience, which is an individual activity, in a reusable format and shares it to improve the quality of the user experience of another individual (Figure 3).

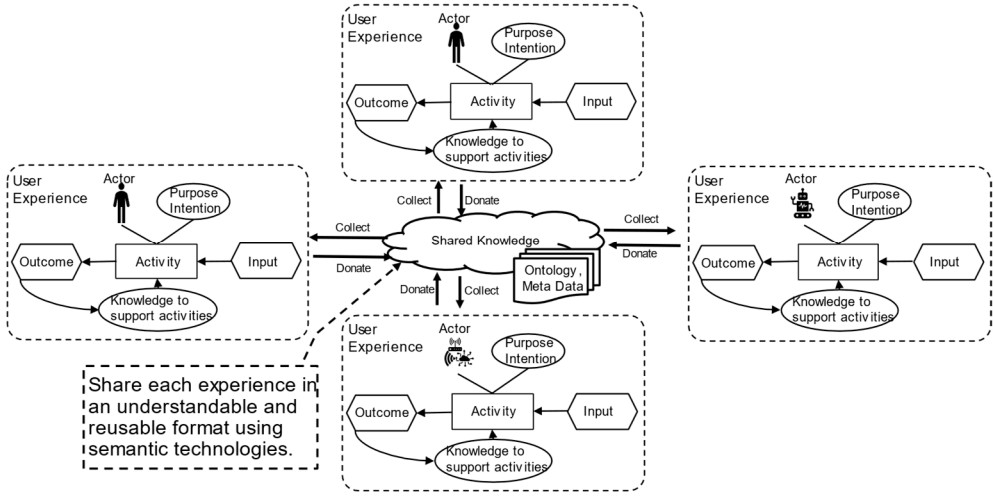

**Figure 3.** Overview of Knowledge Experience.

In the Knowledge Experience, the Actor collects the activities of other related Actors to improve the quality of each activity and donates each experience as knowledge to support the activities of other related actors. To use and share it as knowledge, it is necessary

to record it in a format that each actor can understand and utilize. In the Knowledge Experience, the semantic intelligence technologies, Ontology and Meta Data, are designed and applied to close each gap and enable mutual understanding and utilization.

The following describes the design based on the concept of Knowledge Experience.

Designing based on the concept of Knowledge Experience consists of two steps. In Step 1, the target of systemization is modeled by focusing on input and outcome (deliverables and information) of Actor and Activity (Figure 4).

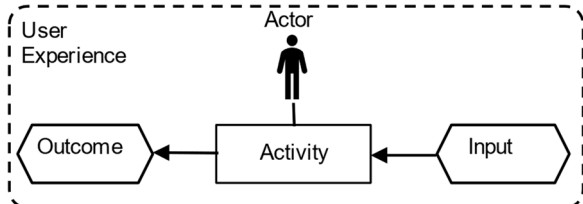

**Figure 4.** Step 1: Focus on Actor, Activity, input, and outcome.

Step 2 adds the following elements to the model in Step 1 (Figure 5): (1) Purpose and intent of the activity; (2) Data that supports the activity; (3) Source of data that supports the activity (User Experience itself).

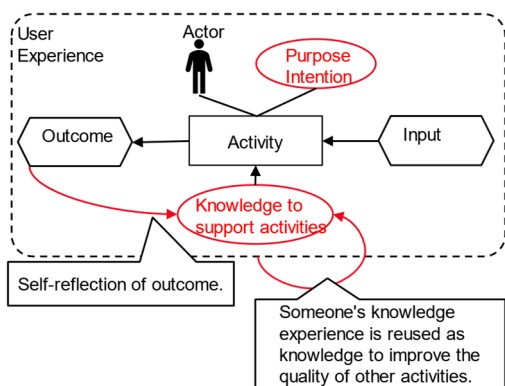

**Figure 5.** Step 2: Add some elements related to the activity.

In an environment based on the concept of Knowledge Experience, knowledge is cyclically and sustainably generated, utilized, and executed as a real activity. In the creation and use of knowledge, it is always a challenge that there is not a small amount of knowledge that cannot be used even if it is shared, and it is known as Inert Knowledge [9]. In response to this, an approach that promotes the use and application of knowledge by sharing "when," "how," and "why" is being practiced in the field of education [10]. In other words, it will be possible to understand and apply the user experience by providing information on the content of the activity and the input and output of the activity, including the purpose and intention of the activity. Specifically, by utilizing the user experience of others as information to support one's own activities, it can be expected that the experience of others will be shared with the group as knowledge and the quality of the group's activities will be improved cyclically and continuously.

Step 1 of the Curator's experience design focuses on the Observe and Orient activities. For Observe, it is necessary to consider the characteristics of museum visitors. The purpose of museum visits varies from person to person but can be broadly classified into (1) Visitors who are willing to spend time watching exhibits for learning (hereinafter referred to as Engaged Visitor); and (2) Visitors who do not expect to spend a lot of time watching the exhibits, for fun (hereinafter referred to as Casual Visitor) [11,12]. And many visitors can be classified as Casual Visitors [13]. In the past, such surveys were mainly conducted using paper questionnaires. However, it was difficult to collect detailed data, especially

from Casual Visitors, because the response behavior was different from the appreciation behavior [13].

There have been several reports of attempts to measure the user experience of visitors in museums by external observation, but most of them adopt VR (Virtual Reality) technology as an exhibition method and record the experience contents in VR [11,14–21], there is almost no measurement report for the conventional physical exhibition appreciation.

In the Smart Museum Project, the viewing behavior of visitors (mainly for Casual Visitors) is externally observed by sensors, and the Curator grasps the user experience of the visitors.

The model obtained from the above considerations is shown in Figure 6.

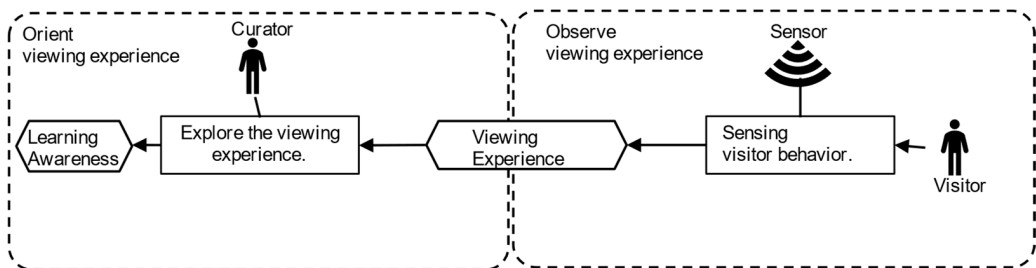

**Figure 6.** Focus on Actor, Activity, input, and outcome in Curator's Experience.

Conventional design techniques that focus on data flow (the relationship between input and output data) focus on ensuring that Sensor output specifications and Curator input requirements are consistent. On the other hand, Knowledge Experience based design focuses on the effect of the data output by the Sensor on the quality of the Curator's activities (contents of awareness). Step 2 derives the design of a mechanism for reusing information as knowledge in order to improve the quality of the Curator's Experience (Figure 7).

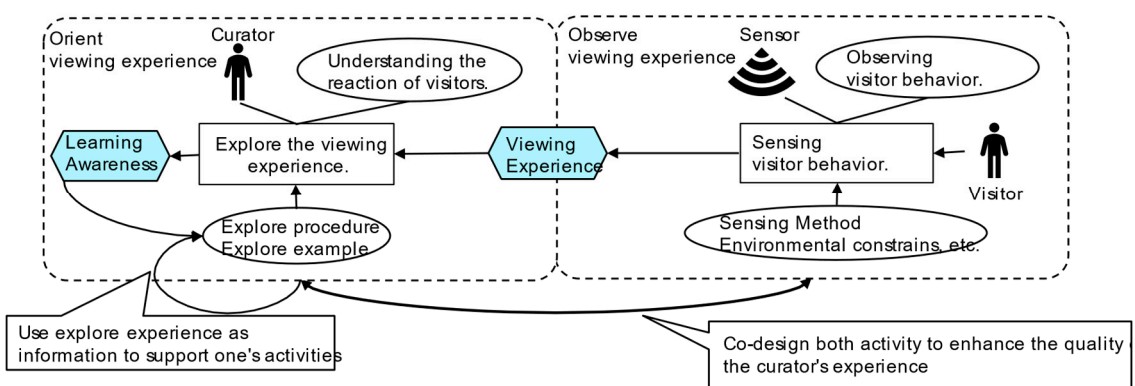

**Figure 7.** Add some elements related to Curator's Experience.

The purpose of the Curator's activities in this model is to understand the reaction of visitors, and it is possible to improve the quality of analysis by utilizing analysis procedures and analysis examples. Analysis procedures and examples can be obtained from the Curator activities that have been carried out so far, and the quality of Curator activities can be evaluated by the learning and awareness that are the result of the activities. From these facts, it can be seen that Curator's activities and Sensor's activities need to be coordinated because Curator's learning and awareness are influenced by the measurement results of visitors by Sensor, which is the input to Curator's activities. In other words, in the design of Sensing visitor behavior, the effect of widening the interval of number measurement is the magnitude of the effect on Curator's learning and awareness. This can be evaluated by the equivalent learning and awareness before and after widening the interval.

### 3. Related Work: Human-Centered Design and Co-Design

The ideas related to the methods presented in this paper are human-centered design and co-design. Human-centered design (ISO9241-210) is the idea of incorporating the user's perspective into the software development process to achieve a usable system [22]. Human-centered design shows the design process and the techniques that can be applied to each process, but the techniques for knowing the user's requirements are general and not shown as specific steps. In the case of the Smart Museum Project, it is necessary to understand the impact of change requests on system components on the user experience and consider the necessary trade-offs. To deal with such cases, a systematic method that incorporates the concept of co-design is required.

The idea of co-design is that the designer makes appropriate design trade-offs for multiple design elements, which is practiced as hardware–software co-design. The hardware–software co-design aims to achieve system-level goals by leveraging the synergies of hardware and software through the simultaneous design of both [23,24]. The idea of hardware–software co-design was mainly targeted at systems on silicon, but efforts are being made to extend it to CPS [24,25]. In co-design for CPS, efforts are being made to coordinate CPS design and CPS service quality, mainly based on objectively (external) measurable results. There is no idea of coordinating the quality of service to end users with the functions that make up CPS from the perspective of user experience, such as human-centered design.

To focus on humans and coordinate systems and user experiences, it is necessary to design proper human–computer collaboration [26]. A prominent problem in human–computer collaboration is that it is difficult for humans to obtain enough information to understand and use the output of computers [27,28]. Communicating instructions and procedures as procedures, rules, and norms is not enough to enable information to guide decisions and actions. It needs to be shared in relation to the underlying grounds, reasons, purposes, and intentions [29]. This discussion is also related to the transparency and explanatory discussion [27,28] of computer calculation results (results of automated learning). Miller's sociological insight into AI's accountability [28] shows that the actions taken are usually explained by goals or intents. It also states the need to explain goals and intentions, as well as calculation results. The same thing has been said in the field of education [10], indicating that an appropriate explanation is required for the recipient to understand and use the information.

The relationship between humans and computers, so far, can be regarded as a battle for initiative between automation by computers and control by humans, and it can be organized that they have created and accumulated information for themselves [26].

In human–computer collaboration, the following items are listed as issues for realizing activities in which humans and computers cooperate: (1) mutual goal understanding; (2) preemptive task co-management (joint management of proactive tasks); (3) shared progress tracking. However, no useful results have been obtained [30].

Introducing initiatives related to human–computer collaboration, user experience is an effort to understand the human perception and response that results from the use of products, systems, and services, as defined in ISO 9241-210. Human–Computer Interaction (HCI) is designed on the assumption that humans can use computers more efficiently. Human-in-the-loop (HitL) is an effort to optimize interpretability by including humans directly in the optimization loop [31]. HitL is designed to involve humans in some decisions and controls in AI systems, centered on machine learning and deep learning [32]. Computer-supported cooperative work (CSCW) is working on how computer systems can be used to support collaboration and coordination [33]. Computer-Mediated Communication (CMC), which is a form of human-to-human communication, via a network computer, has become widespread as a general means of communication, as represented by SNS (Social Networking Service) [34].

The practice of using the information as knowledge to improve the quality of activities has been practiced in the field of Knowledge-Based Engineering (KBE). Verhagen et al. [35]

organized and evaluated KBE-related efforts and presented research topics. KBE is a field of research methodologies and techniques for acquiring and reducing product and process engineering knowledge, to reduce product development time and costs. The issues of KBE are as follows: (1) developers need to identify issues and create individual KBE solutions based on custom development processes (ad hoc development); (2) formulations and captured knowledge formulas and actuals to capture design intent Knowledge cannot be utilized because there is no explanation of meaning and context (black box application).

Domain-Driven Design [36] is a method that incorporates the use of knowledge into software system design. Domain-Driven Design (DDD) can be regarded as a method that incorporates the concept of KBS into object-oriented design, which is a software system design methodology. DDD realizes business activities on the software system by incorporating the business rules in the target business area (activity area) to objects. DDD presupposes that business rules can be determined in advance, but in today's diversified values, it is necessary to adopt business decisions to the varying situations. It is necessary to be able to adjust the rules for making decisions according to the situation. This is an analysis that is consistent with the difficulty of applying rules in environments where uncertainty exists, as described by Cummings [6]. In areas where the process is clear as a business activity and the decision-making criteria can be clarified, efforts are underway to capture and utilize the experience related to decision making as knowledge. More flexibility is needed in cases where the standards of value are not always clear or need to be changed (or may change), depending on the context of the activity [6].

In this way, the cooperation between humans and computers has been considered from the one-sided perspective of either humans or computers. There was no perspective on designing collaboration from both human and computer perspectives and creating, accumulating, and sharing information that would be useful (know, understand, and use) for both humans and computers.

The areas of application for CPS are various (living-related automation, transportation, manufacturing, civil infrastructure, and healthcare) and are becoming more and more related to QOL (Quality of Life) [25]. Along with this, there is an increasing need for a coordinated design of systems and user experiences so that computers and humans can mutually understand and utilize the results of their activities to strike a proper balance between the system and the user experience. As a method for co-design in CPS, there is a need for a method for systematically deriving the connection between the system and the service and the method for assessing the impact on the service when the system components are changed. In other words, it is necessary to extend the conventional concept of human-centered design and the concept of collaborative design to include the quality of the user experience provided by CPS, and to organize the design process for that purpose as a method.

In this paper, the idea of Knowledge Experience is introduced for this issue. Knowledge Experience focuses on the user experience created by the use case and analyzes the activities that make up the use case and the information necessary for the activity. Based on this analysis, co-design is performed at the level of the system (consisting of HW and SW) and service (realized by the system). This method is characterized by coordinating the trade-offs in designing the components of the system at the user experience design level, leading to appropriate design. This makes it possible to derive the relationship and evaluation method between the impact of system design, such as hardware and software and the quality of user experience.

## 4. Implementation of Smart Museum Prototype

The prototype system consists of (1) a sensor agent that collects viewing experiences, (2) a viewing experience repositories, and (3) an environment (Curator's workbench) for Curators to explore and analyze the viewing experience (Figure 8).

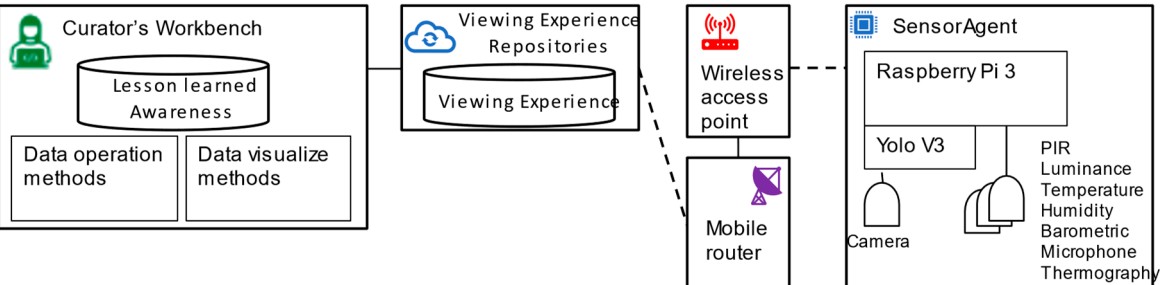

**Figure 8.** Smart Museum Prototype System.

1. Sensor Agent: A sensor network, consisting of multiple sensor nodes, will be introduced in the exhibition room to collect viewing experiences, without relying on active involvement. Individual sensor nodes detect and record visitors who are viewing at that location. It also measures and records the conditions of the viewing environment, such as temperature and humidity at that time. In sensing and recording, in consideration of the right to protect personal data, information, such as images and sounds, that can identify individuals is not saved, and access from the outside is also blocked.
2. Viewing Experience Repositories: Accumulates the viewing experience collected by Sensor Agent and makes it available for Curators to explore and analyze.
3. Curator's Workbench: Explore and analyze the viewing experience using the datasets and methods shared by the Viewing Experience Repositories.

The prototype system was installed into the museum and started trial operation, as shown in Figure 9. Eleven sensor nodes are placed in the exhibition room of the museum and the data from each Sensor Agent are sent to Viewing Experience Repositories, installed at the Data Center via the mobile LTE network, used for exploration and analysis of viewing experience as a knowledge creation activity of Curators.

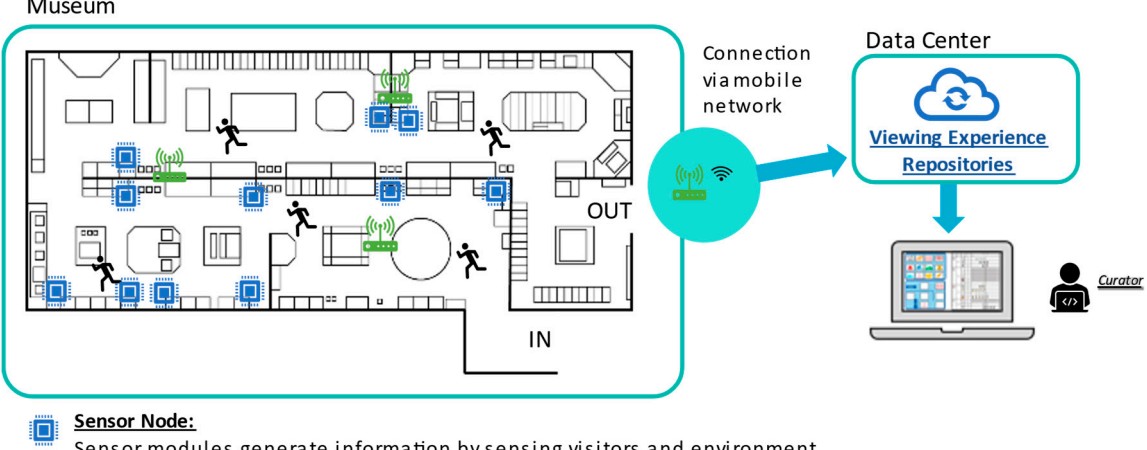

**Figure 9.** Overview of prototype system installed in the museum ([1] Figure 3).

The Sensor Agent is built from a camera, a microphone, a PIR (Passive Infrared Ray) sensor, a temperature/humidity/barometric pressure/illuminance sensor, and a Raspberry PI board. Sensors measure visitor behavior and environmental changes at stationary monitoring points to observe the visitor's behavior. The camera image is processed by the image recognition program Yolo [37], to calculate the number of visitors near the sensor node. PIR senses the movement of people to detect if there are visitors near the Sensor Agent. The microphone measures the volume near the Sensor Agent to determine the presence of visitors. The temperature/humidity/barometric pressure/illuminance sensor is used to record the environmental conditions of the visitor's viewing experience

and to evaluate the impact of changes in the environmental conditions on the viewing experience. Yolo's process of calculating the number of visitors is programmed to run every 60 s. The reason is that, since the installation location is in the museum exhibition room, it is necessary to prevent the sensor board from generating high heat due to the frequent execution of Yolo.

The data measured by the Sensor Node (Table 1) are stored in the Database Server, as a viewing experience for visitors, and is used for exploratory analysis and grasping activities of the viewing experience by the Curator on the Curator workbench.

**Table 1.** Viewing Experience Data (created by modifying [1] TABLE I).

| Name | Device | Intention of Measurement | Measurement-Method |
|---|---|---|---|
| time | Server | Record date and time of measurement | The server records the time when the data is uploaded. |
| agent_serial | Sensor Node | Uniquely identify the sensor agent | The unique number (serial number) of the sensor agent is added at the time of data transmission. |
| temperature | Temperature sensor | Ambient temperature (celsius) | Temperature sensor measurements (measured at 1-min intervals) |
| humidity | Humidity sensor | Abient humidity (%) | Humidity sensor measurements (measured at 1-min intervals) |
| pressure | Barometric pressure sensor | Atmospheric pressure (bar) | Barometric pressure sensor measurements (measured at 1-min intervals) |
| luminance | Illuminance sensor | Ambient luminance (lux) | Illuminance sensor measurements (measured at 1-min intervals) |
| noise.db | Microphone | Ambient noise (dB) | The volume level is recorded by the microphone. (Audio is not collected and conversations are not analyzed or recorded.) |
| motion | PIR motion sensor | Movement of visitors (%) | It is measured once every 0.5 s with a motion sensor to calculate the probability that a visitor was present in the area in 1 min. |
| presence | Camera | Number of visitors | Computer the number of people included in a captured image. (A machine learning algorithm is used to recognize "people", but the individual is not identified.) |

In the Curator's workbench, methods for visualizing data and methods for manipulating data are shared to carry out exploratory analysis and grasp activities of the viewing experience by the Curator. In addition, the awareness gained as a result of the exploration is recorded and shared. In this way, methods and awareness are shared as knowledge to improve the quality of the Curator's activities. This is a design based on the concept of Knowledge Experience.

## 5. Collected Data and Created Knowledge in the Prototype System

Data for evaluation was collected at the Fukushima Museum in Aizuwakamatsu City, Fukushima Prefecture, from 19 June to 2 July 2021.

The Fukushima Museum is open from 9:30 am to 5:00 pm, and there are regularly closed days. The number of visitors during this period as recorded at the admission counter and closed days are shown in Table 2.

**Table 2.** Number of actual visitors during the experiment period.

| Date | | Number of Visitors |
|---|---|---|
| 6/19 | Sat | 83 |
| 6/20 | Sun | 92 |
| 6/21 | Mon | closed day |
| 6/22 | Tue | 135 |
| 6/23 | Wed | 243 |
| 6/24 | Thu | 349 |
| 6/25 | Fri | 338 |
| 6/26 | Sat | 85 |
| 6/27 | Sun | 143 |
| 6/28 | Mon | closed day |
| 6/29 | Tue | closed day |
| 6/30 | Wed | 372 |
| 7/1 | Thu | 356 |
| 7/2 | Fri | no data |

As an example of the data collected, Figures 10 and 11 show the results of measuring the number of visitors on the days with the highest number of visitors (June 30th) and the days with the lowest number of visitors (June 19th). In this example, four measurement points (sensor nodes) are used for simplification. Sensor nodes are numbered and the node numbers and locations, as well as the measurement range for each node, are shown in Figure 12.

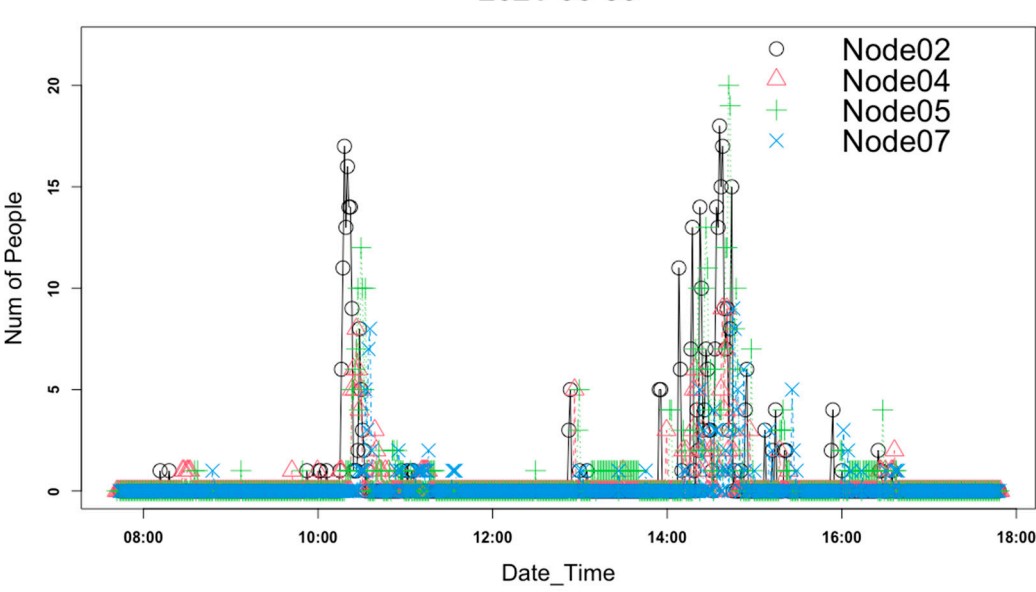

**Figure 10.** Number of visitors detected by the sensor nodes (on 6/30).

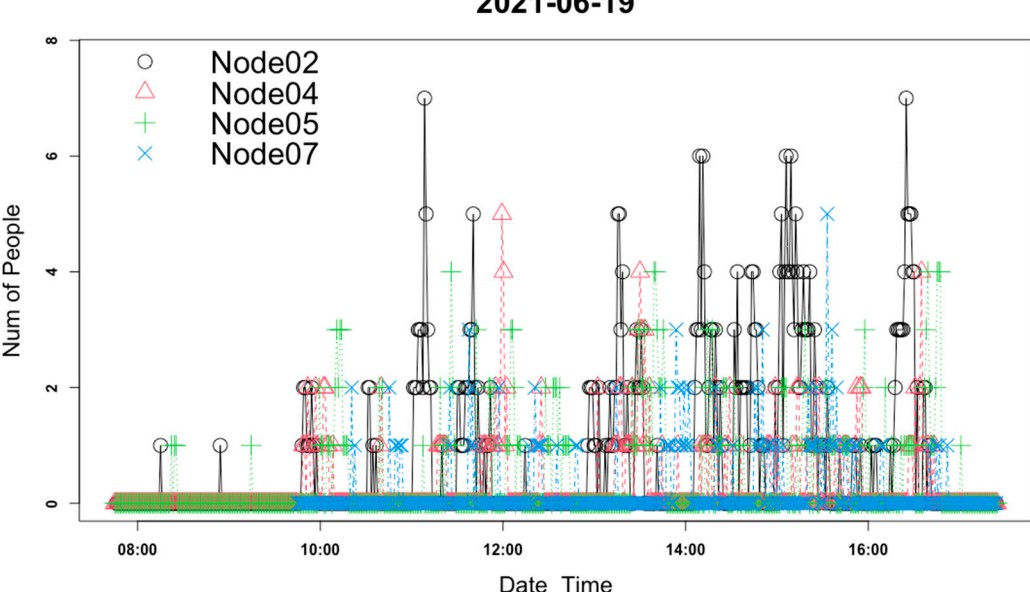

**Figure 11.** Number of visitors detected by the sensor nodes (on 6/19).

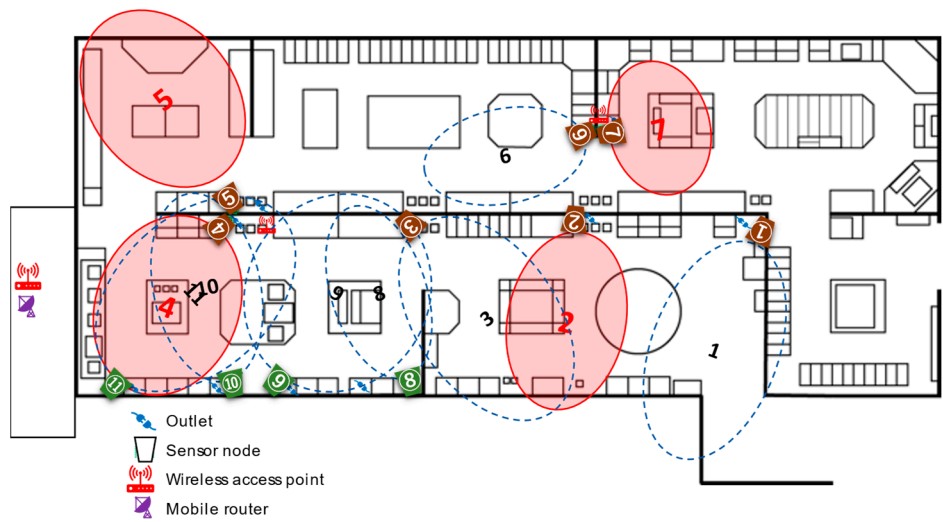

**Figure 12.** Positions of sensor nodes installed at the museum.

The number of people measured depends on the sensor node. The number of people measured by Node02 on both days was generally higher than that of other nodes, which indicates that there is a high level of interest in exhibits near Node02.

The total number of people measured at each sensor node does not match the number of visitors per day (Table 3). This is because there are visitors who were not measured because they passed by without stopping at the place, and visitors who were measured multiple times by the sensor because they stayed at the place to view that area for a longer time. From this data, it is not possible to determine the exact number of visitors who stopped and viewed the exhibits at each location, but it can be interpreted that the visitors who passed by were not interested in the exhibits. In addition, it can be interpreted that visitors who stayed at the place and were measured multiple times are more interested in the exhibits. In this way, by comparing the number of people measured, it is possible to compare the degree of interest of visitors at each measurement location and measurement time.

**Table 3.** The daily total of the actual and detected number of visitors.

| Date | | Number of Visitors | Node02 | Node04 | Node05 | Node07 |
|------|-----|-------------------|--------|--------|--------|--------|
| 6/19 | Sat | 83 | 396 | 150 | 249 | 112 |
| 6/20 | Sun | 92 | 272 | 109 | 272 | 115 |
| 6/21 | Mon | closed day | | | | |
| 6/22 | Tue | 135 | 175 | 51 | 120 | 52 |
| 6/23 | Wed | 243 | 393 | 147 | 297 | 129 |
| 6/24 | Thu | 349 | 622 | 233 | 12 | 230 |
| 6/25 | Fri | 338 | 758 | 275 | 477 | 253 |
| 6/26 | Sat | 85 | 267 | 162 | 152 | 105 |
| 6/27 | Sun | 143 | 419 | 175 | 291 | 179 |
| 6/28 | Mon | closed day | | | | |
| 6/29 | Tue | closed day | | | | |
| 6/30 | Wed | 372 | 423 | 171 | 406 | 141 |
| 7/1 | Thu | 356 | 456 | 146 | 358 | 136 |
| 7/2 | Fri | no data | 225 | 105 | 242 | 82 |

For the Curator to recognize the difference in the level of interest of visitors at each measurement location and measurement time, it is necessary to aggregate the data to some extent. By aggregating the data measured every minute and every hour, it becomes easier to understand the characteristics (Figures 13 and 14).

**Figure 13.** Total value every 60 min (6/30).

Since the placement of sensor nodes were limited by the power soutlet locations in the exhibition room, the sensor nodes could not cover the whole area and also overlaps existed (Figure 12). As has been obtained from similar surveys in the past [38], the Curator's request for the strength of interest is not for each exhibition corner, where multiple exhibits are located, but for individual exhibits. For that purpose, it is necessary to reduce the overlap and omitted area, and be able to identify the object being viewed. In other words, it is necessary to free the placement of the sensor nodes from the limitations imposed by the position of the power outlets.

## 2021-06-19(60min total)

**Figure 14.** Number of visitors detected every 60 min by each sensor node (on 6/19).

### 6. Discussion: Loosening the Constraints on the Sensor Node

Based on the data collection by the prototype system and the evaluation of the knowledge gained from the data, it was found that it was required to enable the installation without the constraint of the Outlet. For that purpose, it is necessary to consider how to reduce the power consumption of the sensor node.

Before the study, the power consumption of the sensor node was measured by the method shown in Table 4.

**Table 4.** Measurement conditions for power consumption.

| Measuring Device | TEXIO PPX 20-5 |
|---|---|
| Measurement method | Measure electric current and voltage at 0.1-s intervals. Calculate the average for every 10 measurements and use the average for 1 s as the measurement result. Measured 300 times (5 min) after stabilized (from about 200 s after power is turned on). |

From the measurement result of the power consumption of the sensor node (Figure 15), it can be seen that the peak occurs at intervals of about 60 s. Since Yolo is programmed to start at the same timing on the sensor node, it is presumed that the peak will occur due to the execution of Yolo. However, in that case, it is necessary to widen the interval for measuring the number of people, but the effect of widening the interval cannot be judged by the sensor node alone, so the system is used. It is necessary to decide the influence on the user (that is, the change in the quality of the Curator's awareness), and it is necessary to have a coordinated design that considers the balance between the two.

Instead of lowering the peak, you can take an approach to reduce power consumption when Yolo is not running by activating the sensor node when Yolo starts up. However, this prototype uses a Raspberry Pi, so it would need to power off and restart. Considering the overhead of powering down, rebooting, and stabilized state, this configuration was determined to be impractical.

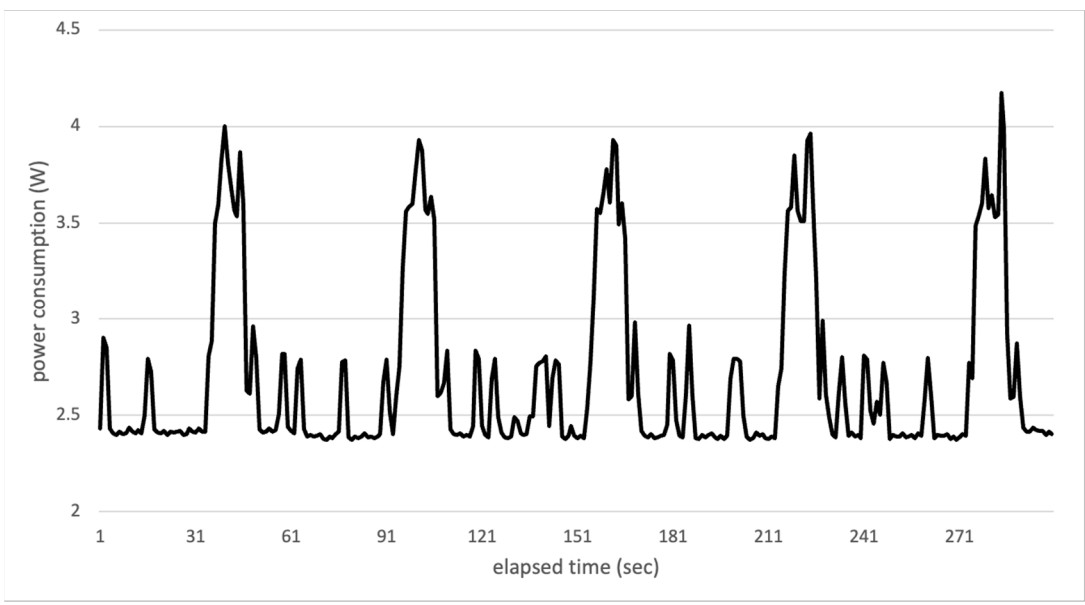

**Figure 15.** Power consumption of a sensor node.

To confirm the relationship between the occurrence of the peak and the execution of Yolo, the programming was changed to start Yolo every 5 min, and the power consumption meter, in that case, was compared. The power consumption measurement conditions are the same as the above conditions (Table 4).

The measurement results are shown in Figure 16. The occurrence of the peak was reduced to one, and it was confirmed that the peak was caused by the execution of Yolo. As a result of this experiment, it was found that the average power consumption is 2.71 W when Yolo is started every minute, while it is 2.51 W when it is started every 5 min. It was also found that the average power consumption can be reduced by 0.2 W, 7.5%, by reducing the number of startups of Yolo to 1/5.

As organized as a model by Knowledge Experience Design, the purpose of observation activities using sensor nodes is to grasp the viewing behavior of visitors, and that knowledge is obtained from the number of people per sensor node, calculated by Yolo. Therefore, it is necessary to judge the validity of changing the activation interval of Yolo by balancing it with the knowledge that the Curator can obtain from the data.

Specifically, the difference in awareness is evaluated by the measured value of the number of people when Yolo is started every minute (1-min value thereafter) and the measured value when the activation interval of Yolo is changed (*n*-minute value thereafter).

The hourly aggregates (hereinafter the 1-h aggregated value) calculated using the 2-min to 10-min values were compared to the 1-h aggregates, based on the 1-min values. The correlation coefficient (R) is as shown in Table 5.

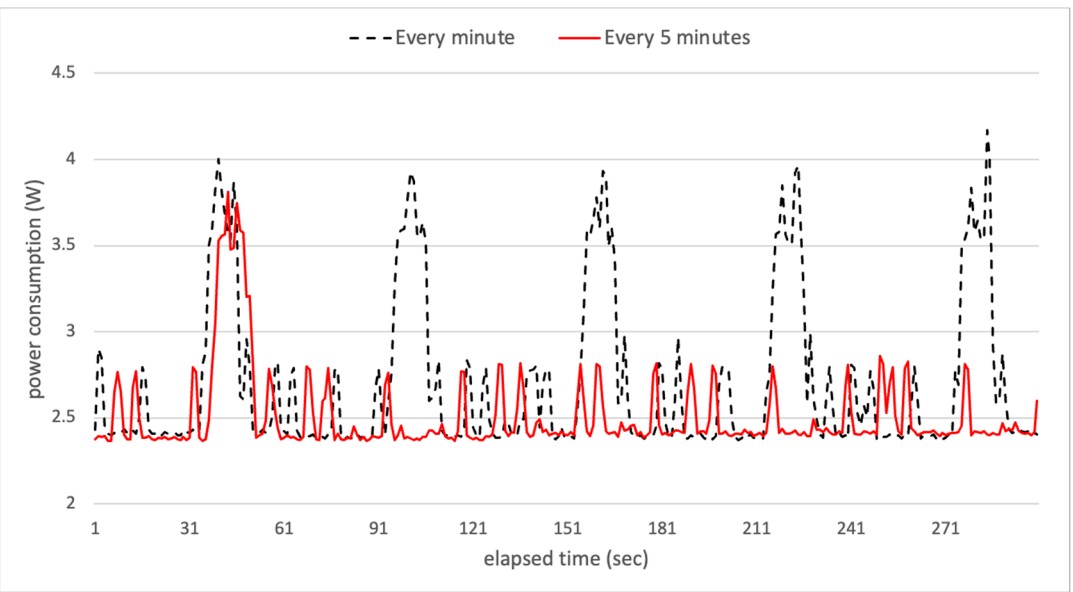

**Figure 16.** Power consumption of sensor node for 1- and 5-min detection intervals.

**Table 5.** Correlation of 1-h aggregate value.

| Yolo Start Interval | 1 min | 2 min | 3 min | 4 min | 5 min | 6 min | 7 min | 8 min | 9 min | 10 min |
|---|---|---|---|---|---|---|---|---|---|---|
| correlation coefficient (R) | 1.000 | 0.990 | 0.980 | 0.962 | 0.937 | 0.933 | 0.917 | 0.895 | 0.879 | 0.837 |
| power consumption (W) | 2.71 | 2.59 | 2.54 | 2.52 | 2.51 | 2.50 | 2.50 | 2.49 | 2.49 | 2.49 |
| reduction rate (%) | 0.00 | 4.61 | 6.15 | 6.92 | 7.38 | 7.79 | 7.91 | 8.07 | 8.20 | 8.30 |

1 min and 5 min are actual measurement values. Other values are estimates interpolated from the 1 min and 5 min values.

The difference in the number of people measured for each time zone is compared using the value obtained by summing up 1-h aggregated value for each day (hereinafter referred to as the daily aggregated value). Figure 17 is a graph of daily aggregated values using the 1-min value, 3-min value, 4-min value, 5-min value, and 7-min value for sensor node 5. As the measurement interval increases, the range of change decreases. Up to the 4-min value, the same tendency as the 1-min value can be grasped, and the range of change between the 5-min value and the 7-min value becomes gradual, making it difficult to grasp the tendency.

Furthermore, the ease of grasping the tendency of the number of people measured for each sensor node was evaluated using the 1-to-7-min values (Figure 18).

In this case as well, as the measurement interval increases, the range of change for each time zone becomes smaller, but up to the 4-min value, the same tendency as the 1-min value can be grasped. However, it is difficult to grasp the tendency of the 5-min and 7-min values because the range of change is gradual.

Judging from this result, if you want to reduce the power consumption of the sensor node by adjusting the startup interval of Yolo, you can judge that it is appropriate to set the startup interval between 4 min and 5 min.

In this way, the knowledge gained by exploratory analysis of the data measured by the sensor node can be expected to improve the measurement method of the sensor node.

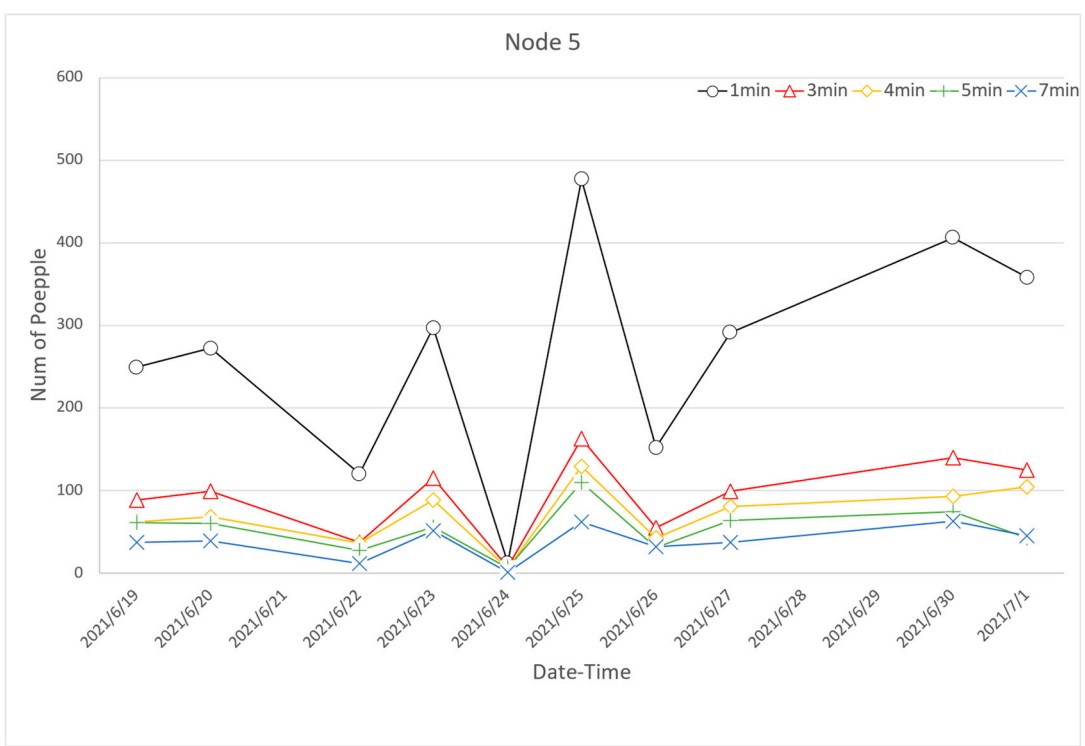

**Figure 17.** Number of visitors detected with different measurement intervals.

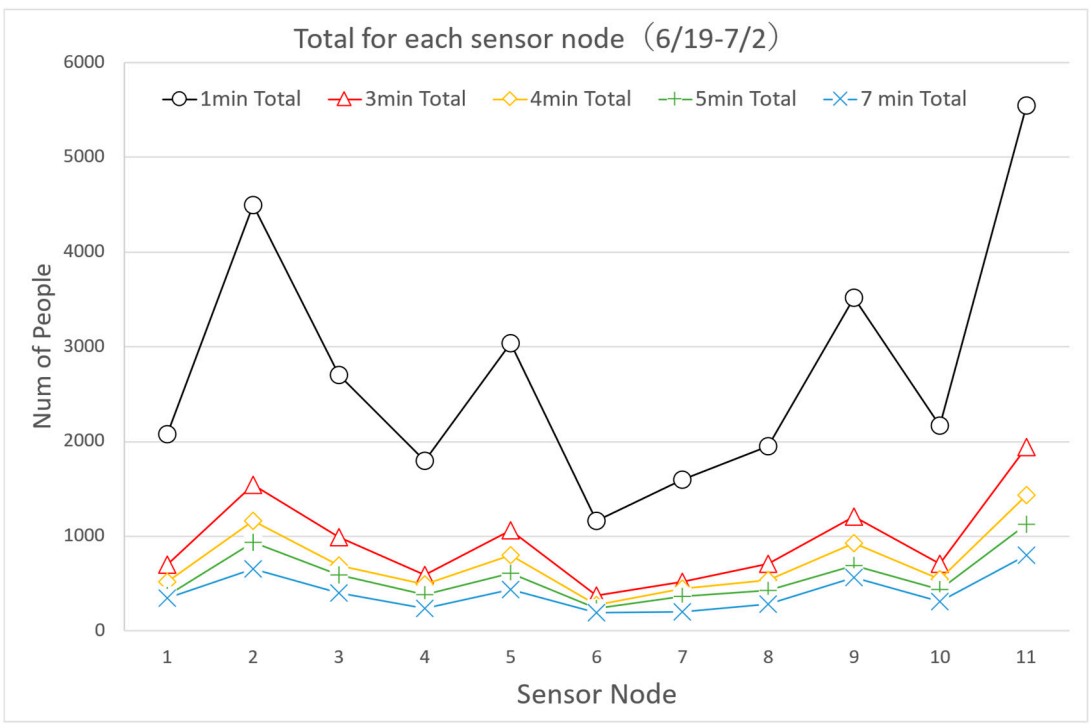

**Figure 18.** The tendency of the number of visitors measured for each sensor node.

### 7. Conclusions

From the prototype system of the Smart Museum, it was found that even if the image analysis program execution interval was changed from 1 min to 5 min to measure the number of visitors, it would not have a significant effect on the Curator's knowledge acquisition. This is useful as a finding on the relationship between the design of system

devices (sensor nodes) and the experience of system users (Curators). With this knowledge, it is possible to reduce the power consumption of the sensor node to 92.6%, by changing the data measurement interval from the 1-min interval to the 5-min interval, without affecting the Curator.

This shows that the knowledge gained by the Curator with exploratory analysis of the data from the sensor node can be used to improve the measurement method of the sensor node. In other words, the Curator's experience and the design of system functions could be coordinated. This is the result of designing the measurement of data by the sensor, not as an interface between physical space and cyberspace, but as an activity (Observe) necessary for knowledge creation by a Curator, and relating it to the purpose and intention of the activity. The method that led to this perspective is Knowledge Experience design.

Thus, an approach that evaluates the quality of the user experience, not just numerical indicators, is effective in designing a form of CPS that provides services to end users.

This achievement can be evaluated not only in terms of reducing the power consumption of the sensor node, while maintaining the quality of the Curator's activities, but also in terms of reducing the amount of data that the sensor node needs to process. By reducing the amount of data processed by the sensor node, it is possible to simplify (Raspberry Pi 3 to Raspberry Pi Zero) the board computer installed in the sensor node. This makes it possible to further reduce the power consumption and size of the sensor node. In addition, further power saving can be expected to be driven by a battery, and the sensor node is not restricted to the position of the outlet in the exhibition room, so it becomes possible to measure the behavior of visitors at any place and measurement range. This will further improve the quality of the Curator's activities.

On the other hand, if it is battery-powered, it is expected that it will be necessary to replace the battery regularly and charge the battery. As future work, it is necessary to conduct experimentsthat combines power consumption and battery capacity and incorporate activities, including system operation, into the Knowledge Experience Design.

In addition, to reduce power consumption approaches, using state-of-the-art electronic devices and intelligent sensors that can manage energy, can be considered [39]. In the prototype system, the sensor node is implemented with the configuration of a relatively inexpensive part that is widely used. In this way, resources, such as budget, are also important considerations for proper co-design, but in Knowledge Experience design, they can be included in the design as constraints included in the purpose and intention of the activity.

In the Smart Museum project, we applied the idea of knowledge experience to hardware–software co-design and tried it as a design method, to optimize the quality of the system (consisting of HW and SW) and the service (achieved by the system). In the case of power saving for the sensor nodes, it was confirmed that it is possible to derive a method to understand and evaluate the effect of changes in system components on services by designing based on Knowledge Experience.

It is expected that a system that optimizes customer service and improves the customer experience by acquiring customer behavior with a device, such as IoT, and analyzing the data will become more familiar as an application example of CPS.

**Author Contributions:** Conceptualization, T.H., R.Y. and Y.K.; methodology, T.H., R.Y. and Y.K.; investigation, all authors; writing—original draft, T.H., R.Y. and Y.K.; visualization, T.H., R.Y. and Y.K. All authors have read and agreed to the published version of the manuscript.

**Funding:** This research received no external funding.

**Data Availability Statement:** Not applicable.

**Acknowledgments:** This research was performed in cooperation with the Fukushima Museum.

**Conflicts of Interest:** The authors declare no conflict of interest.

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
