# Peer review of "Cooperative Design of Devices and Services to Balance Low Power and User Experienceâ€"

_jlpea, doi:10.3390/jlpea12010015_

Round 1
Reviewer 1 Report
In the paper, the authors have presented an extended version of a recent conference paper they have published (Design of a Knowledge Experience based Environment for Museum Data Exploration and Knowledge Creation), now with a focus on the method for evaluating the effect of power consumption of the sensor on the awareness of the museum's curator. Thus, my comments are focused on that particular aspect of the paper.
- The structure of the paper must be improved. The introduction is too long, and a new section on Experimental Design should be create from it.
- No explanation is given on Yolo, and from Figure 4, it is very hard for the reader to understand why is an algorithm as Yolo the main responsible to create power peaks.
- What is the rationale of choosing exactly 1 minute or 5 minutes as references for power measurements? did the authors measure the power consumption below 1 minute interval and checked the benefit or drawback from it?
- What is the influence of the other sensors from Table 1, beside the camera, on the overall power consumption? No useful input data from these sensors were identified as a factor to influence the quality of the curator's awareness.
- From figure 10, the same power floor is clearly visible (2.4 W approximately) for both measurement interval. Thus, it may indicate that the processor is continuously active. could the author consider an interrupt based approach instead of a polling approach in their programming, to improve and reduce the power floor consumption?
- Correlation analysis results should be presented also for 10 and 15 minutes interval measurements.
- Line 371: When considering a battery powered approach, care should be given by the drawback of such approach (regular change and / or charge of the batteries). A rigorous study combining power consumption (W) and battery capacity (mAh) could be a follow-up of this work.
- the conclusions should deepen the power consumption analysis, and its importance in the design of the Knowledge based design.
Other comments:
- The quality of Figures 1, 11, 13, 14 and 15 must be improved. They are pixelated.
- Line 178: SNS acronym should be described
- line 209: ...CPS are various living-related automation... -> ...CPS are various: living-related automation...
- line 249: ...Actor, Activity, and Activity, and model... -> ...Actor, Activity, and model...
Author Response
Dear Reviewer,
Thank you for the thoughtful and constructive feedback you provided regarding our manuscript, (Cooperative design method for systems and services for user experience quality and low power). We agree with your suggestion, and we have amended this by changing the content as on the next page and after.
We are certain that you will find this most recent version of our manuscript clears up the main issues you indicated in your response.
With these changes to our final manuscript, we hereby resubmit our manuscript for a next evaluation. Thank you once again for your consideration of our paper.
Sincerely,
Lead Author:
Takayuki Hoshino (d8202104@u-aizu.ac.jp, takayuki.hoshino@biprogy.com)
Corresponding Author:
Takayuki Hoshino (d8202104@u-aizu.ac.jp, takayuki.hoshino@biprogy.com)
Rentaro Yoshioka(rentaro@u-aizu.ac.jp)

Reviewer 2 Report
This paper explains that adjusting the measurement interval of the sensor can affect the power consumption in the system that monitors the movement of visitors and also provides the information of visitors’ behaviors to the curator in a Smart Museum. Although this paper presents the experimental results for sensor measurement, it lacks a faithful explanation of the specific techniques and procedures for Knowledge Experience Design, especially how each step of the Design method was applied in the experiment.
1. The title of this paper states to suggesting a cooperative design method, but the content of this paper describes only the result of adjusting the sensor measurement interval. It is necessary to explain the procedure for the Knowledge Experience Design method, detailed input/output information, etc. In particular, it is wondering that how the quality of user experience and power consumption are considered in the Design steps.
2. From Table 1,
(1) Among the data collected from sensors, where and how is the environmental information used? It is difficult to understand because there is no specific explanation.
(2) What is the meaning of the words “having people” in the item “Intension of measurement”? Also, what does the term “people (people)” mean?
(3) What does “the probability of detecting movement” mean in “Measurement-Method”? Is it the probability of detecting the viewer's movement or the probability that the sensor will perform a detection action?
3. What are the “Knowledge Templates” mentioned on page 7? It is necessary to explain what information it contains and where it is used.
4. The meaning of the word “Outcome” used in Figures 5 and 6 and “Output” explained in the text is different. It is wondering if the difference between them is clearly used in the paper. It is necessary to explain specifically what the Outcome presented in Figures 5 and 6 means.
5. In Figure 8, are the terms “measurement result” (the output of the sensor) and “Visitor’s Experience,” (the input of the curator’s activity), different? Does the sensor output information change when used as input to the curator activity?
6. This paper mentions “information reuse”,
(1) How is it reused?
(2) In order to be reused, it must have basic meta-data of reusable information. Who creates and manages them? Do you mean feedback rather than reuse?
7. In Section 3. Results;
(1) What is the author trying to claims through this section? Isn't it natural to say that measurement at 5-minute intervals can reduce power consumption than measurement at 1-minute intervals? How about proposing a method to reduce power consumption when performing with the same measurement interval?
(2) Why 5 minutes? Should not it be 4 or 6 minutes? What factors were considered to determine the measurement interval of the sensor? Can you justify that the optimal of sensor measurement interval is 5 minutes, which could affect the curator's activity? If it is 6 minutes, power consumption can be further reduced.
8. It is difficult to know exactly the meaning of what the Figure 11 wants to show. What does the Y axis mean? The circle-symbol, Node02 will mean the sensor node. In this figure, the number of Node02 is about 30 or more. What does this mean? Also, does the red dotted line have a special meaning in the figure?
9. At line 349 on page 10, the correlation is given as 0.98.
(1) This is very difficult to understand. The objects of observation is casual visitors, and 5 minutes means 5-times long period than 1 minute. In the case of measurements taken at 5-minute intervals, although the number of people may be similar, the pattern of movement will be very different. An explanation of the process or movement pattern is required to explain how to get the result of 0.98.
(2) In addition, in order to increase the reliability of the experiment, it is necessary to provide more diverse data. For example, the number of actual visitors, the length of time a person stays in a particular sensing zone, etc.
10. From the overall description of this paper,
(1) This paper states about the designing systems and services, but what are the design procedures and methods, and what are the outputs at each design stage? Although Figures 6 and 7 represent the two steps for design, it is difficult to understand what systems and services are co-designed from the steps. How does the extending the sensing interval to 5 minutes affect to co-design?
(2) What is the difference between your design method and the trade-off analysis in general HW and SW co-design? There are studies that consider power consumption reduction in the existing co-design method.
11. Also, this paper refers to the application in CPS for the proposed method. Can your project, Smart Museum System, be called CPS? CPS should be able to obtain the information from the external physical environment, and intelligently analyze/process it in a computer algorithm, then control the physical environment based the processed information. From that point of view, how or what does the curator's computer control the external exhibition hall or the sensing process?
12. How can it be justified that there are no problems at all in the course of conducting the experiment? Does the machine learning algorithm detect all of visitors accurately in all situations? The threat analysis for the experiment should be explained.
13. The title of this paper states that the purpose of this paper is for low power consumption and quality of user experience.
(1) What activities of the proposed Knowledge Experience Design can contribute to reducing power consumption and providing satisfaction to the user experience?
(2) The value for power consumption is reduced from 2.71W to 2.51W as mentioned on page 10. What does this reduction have an effect on the actual environment? So, did the size of the sensor installed in the museum become smaller?
14. The needs to be corrected in English.
- Case sensitivity (e.g., p.7, Collects ?)
- Use of singular and plural verbs
- Duplicate use of the same word (e.g., p.7, Actor, Activity, and Activity, and..)
- Sentence structures that are difficult to understand (e.g., p.8, focuses on the user experience focuses on the effect of the data output by the …)
- Untying abbreviations (e.g, PIR)
- Use appropriate words (e.g., judgment -> (?) decision)
Author Response

(The authors gave the same response as above.)

Reviewer 3 Report
In this paper, the authors propose the application of the idea of knowledge Experience to hardware-software co-design and tried it as a design method to optimize the quality of the system (consisting of HW and SW) and the service (achieved by the system). The authors also demonstrate that it is possible to derive a method to understand and evaluate the effect of changes in system components on services by designing based on Knowledge Experience in power saving of sensor nodes.
The paper is interesting and easy to understand. However, some major changes are needed in order to improve the scientific soundness and quality of presentation.
- First of all, it is important to highlight that the passive voice or third person is the correct language. Please, change “we” for these forms.
- The image quality of figure 1 is very poor. Please, increase this.
- The fonts in figure 4 and 10 are very small. Please change this.
- On the other hand, it is important to highlight other strategies to reduce the consumption of sensors. For example, state-of-the-art electronics can be used to reduce current consumption [1] or custom and intelligent sensors capable of managing this energy could be used [2].
[1] Castaño, F., Torelli, G., Pérez-Aloe, R., Carrillo, J.M.; Low-voltage rail-to-rail bulk-driven CMFB network with improved gain and bandwidth (2010) 2010 IEEE International Conference on Electronics, Circuits, and Systems, ICECS 2010 - Proceedings, art. no. 5724490, pp. 207-210. DOI: 10.1109/ICECS.2010.5724490.
[2] Castaño, F., Toro, R.M.D., Haber, R.E., Beruvides, G.; Conductance sensing for monitoring micromechanical machining of conductive materials (2015) Sensors and Actuators, A: Physical, 232, DOI: 10.1016/j.sna.2015.05.015.
Author Response

(The authors gave the same response as above.)

Round 2
Reviewer 1 Report
The authors have significantly improved the quality of the paper, by answering to the majority of the questions and following the proposed suggestions for improvements.
Some details must be improved: The quality of the Figures (e.g. 10, 11, 13, 14) should be improved and the overall written text should be revised for clarity.
Author Response
Thank you for the thoughtful and constructive feedback you provided regarding our revised manuscript, (Cooperative design method for systems and services for user experience quality and low power). We agree with your suggestion, and we have amended this by changing the content as on the next page and after.
We are certain that you will find this most recent version of our manuscript clears up the main issues you indicated in your response.
With these changes to our final manuscript, we hereby resubmit our manuscript for a next evaluation. Thank you once again for your consideration of our paper.
Sincerely,
Lead Author:
Takayuki Hoshino (d8202104@u-aizu.ac.jp, takayuki.hoshino@biprogy.com)
Corresponding Author:
Takayuki Hoshino (d8202104@u-aizu.ac.jp, takayuki.hoshino@biprogy.com)
Rentaro Yoshioka(rentaro@u-aizu.ac.jp)

Reviewer 2 Report
The revised paper has many improvements.
1) What does R mean in Table 5? Full sentence is required.
2) Page 15, Line 466: … obtained by totaling the time total value for each day -> Does the term ‘totaling’ mean summing up?
3) Increase the sharpness quality for figures 10, 11, 13, 14, 15, and 16.
4) Please, check word spacing, connector ‘and” usage (e.g., A and B), etc.
Author Response
Dear Reviewer,
Thank you for the thoughtful and constructive feedback you provided regarding our revised manuscript, (Cooperative design method for systems and services for user experience quality and low power). We agree with your suggestion, and we have amended this by changing the content as on the next page and after.
We are certain that you will find this most recent version of our manuscript clears up the main issues you indicated in your response.
With these changes to our final manuscript, we hereby resubmit our manuscript for a next evaluation. Thank you once again for your consideration of our paper.
Sincerely,
Lead Author:
Takayuki Hoshino (d8202104@u-aizu.ac.jp, takayuki.hoshino@biprogy.com)
Corresponding Author:
Takayuki Hoshino (d8202104@u-aizu.ac.jp, takayuki.hoshino@biprogy.com)
Rentaro Yoshioka(rentaro@u-aizu.ac.jp)

Reviewer 3 Report
The authors have modified the article according to the reviewer´s comments. For this, the paper have been improved in terms of quality of presentation and scientific soundness.
Author Response

(The authors gave the same response as above.)
